# The Balance between Different ROS on Tobacco Stigma during Flowering and Its Role in Pollen Germination

**DOI:** 10.3390/plants11070993

**Published:** 2022-04-05

**Authors:** Maria Breygina, Olga Schekaleva, Ekaterina Klimenko, Oksana Luneva

**Affiliations:** 1Department of Plant Physiology, Biological Faculty, Lomonosov Moscow State University, Leninskiye Gory 1-12, 119991 Moscow, Russia; anny.shirly.ganbatte@gmail.com (O.S.); kleo80@yandex.ru (E.K.); 2Department of Biophysics, Biological Faculty, Lomonosov Moscow State University, Leninskiye Gory 1-24, 119991 Moscow, Russia; oluneva@yandex.ru

**Keywords:** pollen germination, plant reproduction, ROS, stigma exudate, SOD, oxygen radicals

## Abstract

The concept of ROS as an important factor controlling pollen germination and tube growth has become generally accepted in the last decade. However, the relationship between various ROS and their significance for the success of in vivo germination and fertilization remained unexplored. For the present study, we collected *Nicotiana tabacum* stigma exudate on different stages of stigma maturity before and after pollination. Electron paramagnetic resonance (EPR) and colorimetric analysis were used to assess levels of O^•^_2_^−^ and H_2_O_2_ on stigma. Superoxide dismutase activity in the stigma tissues at each stage was evaluated zymographically. As the pistil matured, the level of both ROS decreased markedly, while the activity of SOD increased, and, starting from the second stage, the enzyme was represented by two isozymes: Fe SOD and Cu/Zn SOD, which was demonstrated by the in-gel inhibitory analysis. Selective suppression of Cu/Zn SOD activity shifted the ROS balance, which was confirmed by EPR. This shift markedly reduced the rate of pollen germination in vivo and the fertilization efficiency, which was estimated by the seed set. This result showed that hydrogen peroxide is a necessary component of stigma exudate, accelerates germination and ensures successful reproduction. A decrease in O^•^_2_^−^ production due to NADPH oxidase inhibition, although it slowed down germination, did not lead to a noticeable decrease in the seed set. Thus, the role of the superoxide radical can be characterized as less important.

## 1. Introduction

The stigma of a pistil is a small surface with a significant role in angiosperm reproduction: on it, pollen grains land, hydrate and germinate under the control of female sporophyte tissues, which should ultimately lead to successful fertilization [1]. The stigmas of flowering plants can be divided into two main types: dry and wet [2]; wet stigmas are covered with a viscous liquid—the exudate —on which pollen lands. In this work, we focused on the interaction of a male gametophyte with a sporophyte on a wet stigma, since exudate is a liquid that is more or less available for collection and analysis.

Studies of stigma exudate have passed a long historical path from early ideas about exudate as source of water for pollen hydration and storage compounds for sustaining pollen tube growth [3,4] towards understanding the diversity of possible exudate compositions based on metabolic and proteomic approaches [5]. The stigma exudate, as far as we now understand, functionally diverges between species on the basis on their ecology as well as biochemical, morphological and anatomical features of their stigmas. For example, exudate proteins in *Lilium longiflorum* and *Olea europaea* are involved in at least 80 different biological processes and 97 molecular functions, suggesting that this secretion is indeed a biochemically active site [6]. It is interesting to point out that the lily and olive stigma exudates only share about 22–25% of the identified proteins. In contrast to proteins, the content of which is as specified, stigma exudate also contains quite universal components, such as ROS [7,8]. ROS are synthesized on stigmas of various (if not all) flowering plants; this property is common for all species studied to date including *Magnolia* [9,10,11,12]. Moderate activity of some ROS-regulating enzymes, in particular peroxidases [13,14,15], has been shown in several species from *Asteracea* and *Solanaceae*; in sunflower, SOD activity was also visualized [15]. On the other hand, pollen exine and pollen suspension have been found to have antioxidant properties [16,17]. Thus, the final balance between ROS production and elimination during in vivo germination is the result of a complex interaction between sporophyte and male gametophyte, which includes both low molecular weight components and antioxidant enzymes [18].

In accordance with the current concept of exudate as a medium for interaction between pollen and stigma [5], a hypothesis was put forward that one of the functions of ROS on the stigma is to support and/or stimulate pollen germination. This hypothesis was based, on one hand, on data on physiological effects of ROS on pollen tubes in vitro [19,20], and on the other hand, on ROS in the stigma exudate [7,8]. It turned out that hydrogen peroxide is present in tobacco stigma exudate in significant amounts. It causes membrane hyperpolarization in pollen tubes and protoplasts; treatment with catalase or ROS quencher MnTMPP abolishes this effect [17]. However, not only hydrogen peroxide can affect the male gametophyte. Recent work has shown that superoxide radical generation in vitro resulted in a marked acceleration of lily pollen tube growth and other physiological effects [21]. Considering that the main source of extracellular ROS on the stigma, as in other tissues, is most likely plasmalemma NADPH oxidase [22], probably both ROS are present on the stigma: superoxide and peroxide. To date, the question of the relationship between various ROS and their significance for the success of in vivo germination and successful fertilization has remained open.

For the present study, we chose a well-studied object with a wet stigma, *Nicotiana tabacum*, identified four stages of stigma maturity before (1–3) and after (4) pollination, and assessed the level of superoxide and peroxide production as well as SOD activity at each stage. In addition, using inhibitory analysis, we established the contribution of redox metabolism enzymes to the successful pollen germination in vivo.

## 2. Results

### 2.1. ROS Dynamics on Stigma

For the convenience of studying ROS production dynamics, we conditionally subdivided the flower development into 3 stages before pollination and one after pollination. The first stage corresponded to the initial coloring of the corolla. The second stage corresponds to corolla opening, and the third to dehiscence of anthers and the change of corolla color to pink (Figure 1a). Stage 4 corresponds to pistils the next day after pollination, thus they had already lost their susceptibility to pollination. We did not study early stages of flower development, since the size of the stigma was significantly different from the mature one, and the style was too soft for the experiments—our research was focused on the stages directly preceding maturity and maturity itself.

We have used EPR spectroscopy for the first time to evaluate the production of the superoxide radical on the stigma during flower development and pollination. As oxygen radicals are unstable, we used a DEPMPO spin trap to specifically collect O^•^_2_^−^ produced during pistil incubation. Incubation of stigmas with the spin trap was performed on intact flowers of maturity stages from 1 to 4 on growing plants. 

Juvenile stigmas produced the superoxide radical much more actively than mature ones (Figure 1a,b). The signal at each of the stages 2, 3 and 4 was significantly different from that of the previous stage. The highest level of O^•^_2_^−^ was detected at stage 1, the lowest at stage 3. After pollination, superoxide radical production increased again, reaching approximately the level of stage 2. It should be taken into account that we removed the pollen that got into the exudate collected at stage 4 so that it could not affect the signal level.

The level of hydrogen peroxide, which we estimated in the fresh exudate spectrophotometrically, was also at its maximum on juvenile stigmas and reached a minimum in pollinated flowers; however, in this case, the signal level at stage 4 was significantly lower than before pollination (Figure 1c). It should be noted that the level of peroxide at each of the stages was very stable in many experiments; the values did not overlap (which can be seen from very short error bars).

### 2.2. SOD Activity Dynamics and Isoenzyme Composition

The conversion of superoxide to hydrogen peroxide is catalyzed by superoxide dismutase (SOD), and this reaction is thought to be one of the main ways H_2_O_2_ is produced on the stigma. To understand how active this pathway is in juvenile and mature flowers and how its activity changes over time, we used the zymographic method for determining SOD activity. Gels of two densities were used to ensure the reproducibility of the results. Both 10% and 15% gel showed two bands, apparently corresponding to two SOD isozymes. The only difference is in the position of the bands on the gel—in the 10% gel they go further due to the lower density (Figure 2a,b).

Both gels show that the activity of SOD at the first stage is lower than at subsequent ones, and the second isozyme is practically not detected in juvenile flowers (Figure 2a). The level of activity of both isozymes at stages 2 and 3 are the same. At stage 4, the results varied greatly: in some cases, high activity was detected (as on the gel in Figure 2a), and in others, it was about the same as in stages 2 and 3 (as on the gel in Figure 2b). Either both isozymes or only one were detected in stage 4. We do not know what caused this, and we do not draw any conclusion on this matter.

There are three distinct types of SODs: Cu/Zn SOD, MnSOD and Fe SOD. Different sensitivities of SOD types to specific inhibitors (hydrogen peroxide and potassium cyanide) make it possible to distinguish them. To determine the type of SOD responsible for each of the SOD activity bands from tobacco stigma, native PAGE gels were treated with different inhibitors before activity assessment. KCN causes inactivation of Cu/Zn SODs and shows the activity of Mn SODs and Fe SODs. Of the two bands present in the control extracts, the lower one disappeared and the upper one remained (Figure 2c). Cu/Zn SODs and Fe SODs are sensitive to H_2_O_2_, so pretreatment H_2_O_2_ can identify Mn SOD. We found that this type of enzyme does not contribute to total SOD activity in tobacco stigma, as no activity was determined after incubation with H_2_O_2_ (Figure 2d). Thus, two types of isoenzymes are responsible for SOD activity on tobacco stigma: one Cu/Zn SOD and one Fe SOD. Since the lower band in the extracts of the first stage was very weak or completely absent (Figure 2a,b), we can conclude that Cu/Zn SOD appears later during stigma ontogenesis.

### 2.3. Significance of ROS Balance for Pollen Germination In Vivo

The ultimate goal of pollination and pollen germination is successful fertilization. Both the percentage of pollen germination on the stigma and the rate of tube growth can affect its effectiveness. To reveal the significance of reactive oxygen species detected on the stigma, we shifted ROS balance with inhibitors. As we have shown, SOD on the stigma is sensitive to hydrogen peroxide and potassium cyanide, but we could not apply these inhibitors to the stigma since the peroxide itself is a component of the redox system and cyanide is toxic to living tissues. Therefore, for experiments in vivo, we used only a specific Cu/Zn SOD inhibitor LCS1. We were able to test the effectiveness of the inhibitor directly on the stigma: the O^•^_2_^−^ content in the exudate on LCS1-treated stigmas increased by 38% (Figure 3a,b).

By preincubating stigmas with this inhibitor, as well as with two inhibitors of NADPH oxidase, likely one of the key ROS producers on stigma, we assessed changes in the rate of pollen germination on stigma and fertilization efficiency. NADP-oxidase inhibitors had an effect on the germination rate: DPI and acetovanillone (AV) reduced the number of tubes in the style by 59 and 76%, respectively (Figure 3c,d). However, when we tested the most effective inhibitor in this regard, AV, in terms of fertilization success, no significant reduction was found (Figure 3e). Apparently, the number of tubes that eventually grew to the ovary was enough to fertilize a sufficient number of ovules. Thus, the reduction in O^•^_2_^−^ formation affects the dynamics of germination, but not the efficiency of fertilization.

On the other hand, a shift of ROS balance towards the superoxide radical (by blocking one of the SOD types) significantly reduced the germination rate: after half an hour, there were no pollen tubes in the style (Figure 3c,d). SOD inhibition also had an effect on the fertilization efficiency: the number of seeds in a capsule was 31% less than in control (Figure 3e), i.e., this parameter changed in proportion to the effect of LCS1 on O^•^_2_¯ content (Figure 3a,b). During the experiment, we noticed that that capsules formed from LCS-treated flowers matured more slowly than those formed from control and AV-treated flowers: we had to wait an additional week until they were completely dry.

## 3. Discussion

Pollen tube germination, growth and guidance throughout the pistil tissues is a tightly regulated process that comprises a continuous exchange of signals, both physical and chemical [1,23]. It has been shown that stigmas from different species accumulated ROS when they are receptive and that these levels decrease on stigmas supporting pollen development [24]. The idea that ROS accumulation goes hand in hand with pistil maturation and corresponds to preparation for pollination seems to be the most understandable and easily explained. However, a study conducted on a large number of species showed that this pattern is true only for some species, while in others, on the contrary, the amount of ROS decreases before pollination [10]. However, the fact that ROS are produced in the maturing pistil is common to all flowering plants studied to date [8,9,20]. 

We used a set of methods to evaluate the dynamics of superoxide radical and hydrogen peroxide production during stigma maturation, as well as the activity of the enzyme, which catalyzes the transition from one form to another. Previous studies have mostly focused on H_2_O_2_, since it is the most stable ROS and presumably the most important in terms of physiological activity [12,25,26]. We used EPR spectroscopy to evaluate O^•^_2_^−^ production on stigma during flower development and pollination, which made it possible to compare the obtained dynamics with the one described previously for developing olive flowers, where dyes were used for ROS detection [27]. The results are partly consistent with those conducted on olive flowers [27]: juvenile stigmas produced much more ROS than mature ones; however, in olive, the gradient decrease has been only reported for H_2_O_2_, while in tobacco it was observed for the superoxide radical as well. A similar pattern of total ROS production has been previously observed in some studied plants, for example, in magnolia, poppy and pea: the amount of ROS decreased as the flower matured [10]. However, in other plants, the amount of ROS, on the contrary, increased [10], which we also recently observed by conducting similar experiments on lily (unpublished results). It is not yet clear what exactly determines these opposite patterns of ROS dynamics: whether the systematic position of the plant, its ecological niche, or structural features of the flower are important. One of the possible explanations is the species specificity of metabolites with antioxidant activity. Thus, the hypotheses concerning the competition between ROS and flavonoids was tested in ornamental kale (*Brassica oleracea* var. *acephala*): flavonoid level declines as the stigma reaches maturity, which results in ROS accumulation and allows compatible pollination to occur [28]. Moreover, the decreased ROS levels after stigma treatment with exogenous flavonoid (kaempferol) drastically reduced pollen attachment and germination [26]. On the other hand, the tomato mutant with reduced flavonol production *anthocyanin reduced* was characterized by an increased ROS level in pollen and its reduced germination in vitro and in vivo [25]. This work also emphasizes the relationship between reproduction and stress: the increase in ROS under conditions of temperature stress was also more pronounced in the mutant.

Anomalous ROS homeostasis is one of the most important mechanisms of inhibition of sexual plant reproduction, including both reproductive development and fertilization, under high-temperature stress, salt stress and drought [26,29,30,31]. Thus, comparing the production of various ROS on the stigma of stress-resistant and sensitive wheat varieties, the authors came to the conclusion that in the second case, the production was noticeably increased under conditions of complex stress [32]. These results show that although a local increase in ROS level on stigma is important for reproductive success, an excessive increase may indicate stress and inhibit reproduction.

Another important factor controlling ROS balance on stigma is the activity of antioxidant enzymes such as SOD and peroxidase. ROS scavenging has been widely correlated with launching of stigmatic receptivity by means of the increased activity of these enzymes, including the appearance of new isozymes during stigma morphogenesis [14,15]. For example, in sunflower, an increase in total SOD activity was attributed to the staminate stage and was followed by a peak of peroxidase activity during pistillate stage [15]. Tests for peroxidase activity have become the common method to measure pistil receptivity [33]. In ornamental kale, a heatmap generated for 17 proteins with oxidoreductase activity based on their relative abundance at different developmental stages showed a gradual upregulation of these proteins during stigma development [26]. In fact, this is what we observed in extracts from tobacco stigmas: SOD activity is lowest at the juvenile pistil stage; at stages 2 and 3, when the flower opens and exudate is produced, it is at its maximum, and at this time, an additional isozyme appears. The increase in SOD activity and the number of isozymes can be logically associated with the observed decrease in the level of superoxide radical in a mature pistil.

After pollination, SOD activity varies greatly for Fe isozyme (it can be either lower or higher than at the peak of receptivity), and it is consistently lower for Cu/Zn isozyme. The reason for this variability remains to be elucidated. Since we took pistils pollinated the day before, but not strictly 24 h before, for the study, we preliminarily associate the scatter of values with the dynamics of stigma aging, which we did not study in this work.

Different functions have been speculated for ROS on stigma: loosening of cell wall components in order to allow the penetration of pollen tubes, defense against pathogens and pollen–pistil communication [12,27,28]. ROS are likely required for cell expansion during the stigma morphogenesis, as has been widely reported for other organs [34,35]. High levels of ROS may be the result of synthetic activity and active metabolism in stigma cells and surrounding tissues [25], as stigma maturation is accompanied by the change in cell wall rigidity and the launch of exudate production. The idea of ROS accumulation as a protective mechanism against pathogen attack, on the same basis as in flower nectars, has been proposed by Hiscock et al. [12]. This explanation seems to be primarily suited to open flowers with a sugary nutrient exudate, while in tobacco, the accumulation of ROS precedes the appearance of visible droplets of exudate and flower opening. On the other hand, tobacco stigma exudate contains large amounts of lipids, carbohydrates and other components [36], which makes the hypothesis of ROS as markers of synthetic activity quite plausible. The relatively low level of ROS at the fertility stage that we observed, however, was important for pollination, as shown by experiments with inhibitors, so we also consider pollen–stigma interactions to be an important function of ROS. An increase in O^•^_2_¯ production after pollination may be associated with degradation of the upper parts of the pistil, which are not needed for the formation of fruit and seeds. The simultaneous decrease in the H_2_O_2_ level indicates that this superoxide production is not related to pollen–stigma interactions.

The decrease in peroxide levels after pollination is in accordance with the current hypothesis that the adherence of pollen grains triggers the decline of ROS concentration on stigma, probably through nitric oxide (NO) release from pollen [13,16,28]. In addition, the decrease in the ROS level on the stigma after pollen attachment is important for pollen hydration [37]. A recent study demonstrated that perception of pollen coat protein-Bs (PCP-Bs) by ANJEA-FERONIA (ANJ-FER) receptor kinase complex is required for pollen hydration [38]. In *Arabidopsis*, when pollen lands on the stigma, PCP-Bs inhibit the interaction of RALFs (rapid alkalinization factors) and ANJ-FER, and the subsequent decline of stigmatic ROS production allows hydration and opens the gates to pollen germination [38]. Therefore, a possible scenario is that a certain level of ROS on the mature stigma is favorable for the early stage of the pollen–stigma interaction, and a further decrease in ROS after pollen landing might support pollen tube growth in the stigma tissue [8,37]. Based on our data, we can clarify that this reduction is mainly attributable to peroxide, while O^•^_2_¯ level can remain high and even increase.

Since we observe two ROS on the stigma, one of which turns into the other, an important question is which of them causes physiological effects. In vitro experiments have shown that both hydrogen peroxide and superoxide have diverse physiological effects on pollen tubes [18,22,23]. However, which of them are realized in vivo?

We compared the in vivo effect of different inhibitors, one of which directly blocks the production of superoxide and, presumably, indirectly reduces the level of H_2_O_2_, and the other, on the contrary, directly reduces the level of peroxide and increases the level of O^•^_2_^−^. Since the balance between O^•^_2_^−^ and H_2_O_2_ is regulated mainly by the SOD [39], we shifted it by inhibiting one of the isozymes of the enzyme. Half an hour later, not a single tube was found in the style, and a few weeks later, the number of seeds in the capsule was noticeably reduced. Since, in mature pistils, SOD is represented by two isozymes, and we blocked only one of them, it is clear that such an effect could not completely block the growth of tubes and fertilization, but even so, we observed a decrease in the rate and efficiency of germination in vivo. This means that, first of all, hydrogen peroxide is crucial for normal pollen germination, and O^•^_2_^−^ cannot replace it. These data are in good agreement with the result previously obtained in vitro, where the effect of the stigma exudate on tobacco pollen protoplasts disappeared after its treatment with catalase [17].

However, blocking another enzyme of redox metabolism, NADPH oxidase, with two different inhibitors, while reducing germination rate, did not ultimately reduce seed production, which was quite surprising. However, explaining the obtained results, we put forward an alternative hypothesis that, under suppression of the NADPH-dependent pathway, the NADPH-independent pathway of H_2_O_2_ formation is realized, for example, through polyamine oxidase. The hypothesis that polyamines play an important role in pollen germination and tube growth through the regulation of redox homeostasis has been previously tested [40,41], but the issue of polyamines as a source of ROS on the stigma remains to be explored.

## 4. Materials and Methods 

### 4.1. Plant Cultivation, Stigma Exudate Collection and Pollination In Vivo

Plants of *Nicotiana tabacum* L. var. Petit Havana SR1 were grown in a climatic chamber in controlled conditions (25 °C, 16 h light) in vermiculite. The plants were watered with salt solutions. Anthers were removed just before flower opening and dried at 25 °C for three days. Mature pollen was collected and preserved at −20 °C as a standard pollination material.

Stigma maturity was assessed according to flower appearance and was divided into 4 stages (Figure 1a), where stage 3 is a fully mature unpollinated stigma and stage 4, a pollinated stigma (the next day after pollination). Exudate was collected from all stigmas by a “cap method” (Figure 4): a pipette tip containing 10 μL of distilled water and/or spin trap solution was put on the pistil and incubated for 30 min (H_2_O_2_ assessment) or 1 h (with a spin trap for EPR). Then the tip containing the drop was carefully removed, and drops from different flowers of the same stage were placed in an Eppendorf tube and analyzed immediately.

Inhibitors of SOD LCS-1 (1,4,5-dichloro-2-m-tolylpyridazin-3(2H)-one, 4,5-dichloro-2-(3-methylphenyl)-3(2H)-pyridazinone) (Calbiochem) or NADPH-oxidase DPI (diphenyleneiodonium chloride) (Sigma) at final concentrations of 10 µM and 200 µM, respectively, were added to pistils 20 min before exudate collection or pollination in a 1–2 µL drop of water.

For controlled pollination, a standard sample of defrosted pollen (1.5 mg) was added to a receptive pistil treated with inhibitor or equal volume of water and evenly distributed over it with a spatula. Germination was evaluated after 30 min (styles were fixated in 70% ethanol for 10 min); seed set was evaluated after 3–4 weeks, when the fruits were completely dry.

### 4.2. Evaluation of Pollen Germination In Vivo and Fertilization Success (Seed Set) 

In vivo germination rates were assessed on longitudinal sections of styles stained with decolorized aniline blue (0.1% (*w*/*v*) aniline blue in 108 mM K_3_PO_4_) for 30 min to visualize pollen tube callose [42].

The seed set was evaluated by weighing the entire mass of seeds removed from the capsule. Calibration, carried out on numerous samples, allowed us to establish a linear correlation between the total weight and the number of seeds (1 seed = 55 µg).

### 4.3. EPR Spectroscopy

Electron paramagnetic resonance (EPR) spectroscopy is a highly sensitive analytical technique for studying free radicals [43]. The physical phenomenon underlying EPR spectroscopy is the interaction between an external magnetic field and the magnetic moment of an unpaired electron. Spin traps are used to react covalently with the radicals and form a more stable adduct which can be measured [44]. Superoxide radical generation was assessed by EPR in experiments with spin trap DEPMPO (5-(Diethoxyphosphoryl)-5-methyl-1-pyrroline-N-oxide, 2-Diethylphosphono-2-methyl-3,4-dihydro-2*H*-pyrrole1-oxide, (2-Methyl-3,4-dihydro-1-oxide-2*H*-pyrrol-2-yl)diethylphosphonate) (Sigma, Darmstadt, Germany) specific to O^•^_2_^−^. The spectra were recorded at room temperature with RE-1307 spectrometer (Moscow, Russia) operating at microwave power 20 mW, frequency 9.5 GHz, 1 min of sweep time. Each characteristic spectrum is the sum of 20 signal accumulations.

### 4.4. Spectrophotometry

H_2_O_2_ in stigma exudate was detected with the FOX-1 method [45], which is based on the peroxide-mediated oxidation of Fe^2+^, followed by the reaction of Fe^3+^ with xylenol orange. To assess H_2_O_2_ level, stigma exudate was diluted and mixed with equal volume of freshly prepared assay reagent (500 µM (NH₄)₂Fe(SO₄)₂·6H₂O, 50 mM H_2_SO_4_, 200 µM xylenol orange in 200 mM sorbitol). Absorbance of the Fe^3+^-xylenol orange complex (A_560_) was detected after 5 min. Samples were measured with SmartSpec spectrophotometer (BioRad, Hercules, CA, USA).

### 4.5. Zymographic Detection of SOD Activity

Superoxide dismutase (EC 1.5.1.1) activity was detected according to Sharma and Bhatla [15], with modifications. An amount of 200 mg of stigmas were collected from fresh flowers on each developmental stage and homogenized at 0 °C in 0.5 mL of Tris–HCl buffer (50 mM, pH 7.0) containing 50 mM NaCl, 0.05% Tween-20 and 0.1% protein inhibitor cocktail (Sigma, Darmstadt, Germany). The homogenates were centrifuged at 10,000 g, 4 °C, for 20 min, and the supernatants were used for estimating SOD activity. 20 µg of total soluble protein (volume was corrected according to Bradford measurement results) from each homogenate was mixed with non-reducing Laemmli sample buffer and loaded on 10 or 15% PAAG. Vertical gel electrophoresis was performed at 180 V for 2 h at 4 °C. Half of the gel was washed and subsequently soaked in 0.5 mM nitro blue tetrazolium (Roche, Penzberg, Germany) in the dark. After 30 min, the gel was transferred to 50 mM phosphate buffer (pH 7.8) containing 28 µM riboflavin and 28 mM TEMED, incubated for 20 min with gentle agitation and then exposed to light at room temperature. During illumination, the gel became dark blue except at positions containing SOD activity. Illumination was stopped when maximum contrast between achromatic zone and blue color was achieved. To test the efficiency of protein alignment of the samples, the other half of the gel with the same samples was stained with 0.1% Coomassie G250.

### 4.6. Statistical Analysis

Experiments were performed in four to seven independent replications. Results in the text and figures, except the original pictures, are presented as means ± standard errors. Significant difference was evaluated according to Student’s *t*-test. (* *p* < 0.05, ** *p* < 0.01).

## Figures and Tables

**Figure 1 plants-11-00993-f001:**
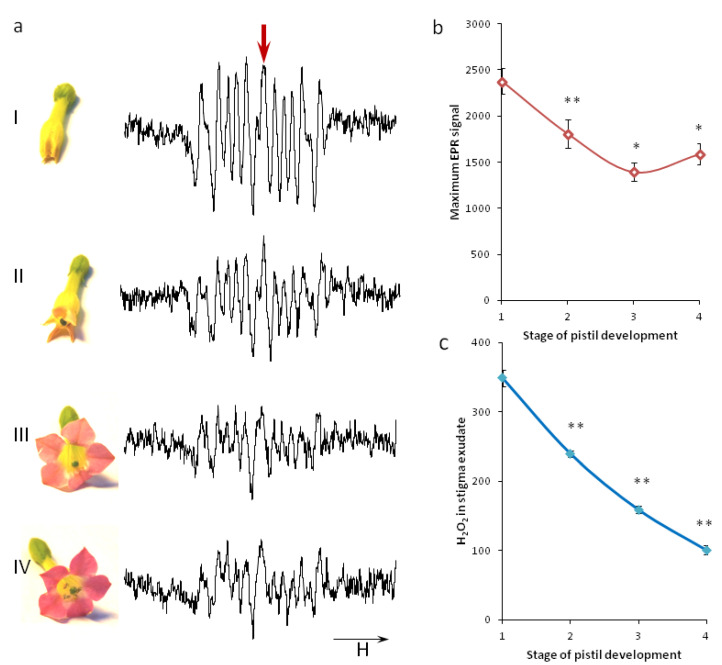
ROS dynamics on tobacco stigma: (**a**) stages of flower development and corresponding typical EPR signals of stigma exudate; (**I**–**III**) unpollinated flowers; (**IV**) pollinated flower (24 h after pollination); (**b**) averaged values of EPR signals (the second peak, indicated by an arrow in **a**), n = 6 (**I**), 7 (**III**), 9 (**II**,**IV**); (**c**) H_2_O_2_ in stigma exudate assessed by FOX-1 method, n = 5 (**I**), 8 (**II**,**III**), 7 (**IV**). The direction of the magnetic field sweep is shown by an arrow below the spectra. Each exudate aliquot (n) was collected from 25–40 (EPR), 15–25 (spectrophotometry) flowers of the same stage. Asterisks indicate a significant difference between the means (* - *p* <0,05, **- *p* <0,01).

**Figure 2 plants-11-00993-f002:**
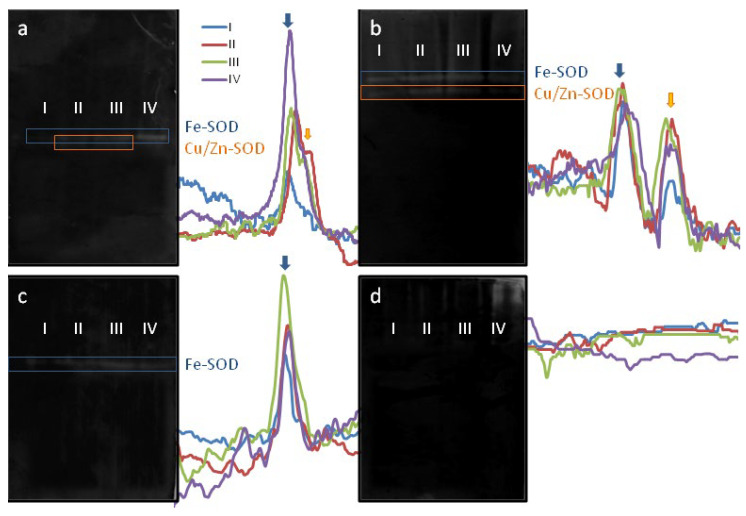
Zymographic detection of superoxide dismutase (SOD activity) and identification of isoenzymes in tobacco stigma. Total protein extracts from fresh stigmas of flowers were loaded into adjacent lanes. Prior to loading, protein concentrations in all extracts were determined by Bradford to adjust the application dose. After native PAGE, the following treatments were conducted, and then gels were stained with NBT reduction: (**a**) no treatment, 10% gel; (**b**) no treatment, 15% gel; (**c**) 3 mM KCN, 10% gel; (**d**) 5 mM H_2_O_2_, 15% gel. Optical density profiles are shown next to the gel.

**Figure 3 plants-11-00993-f003:**
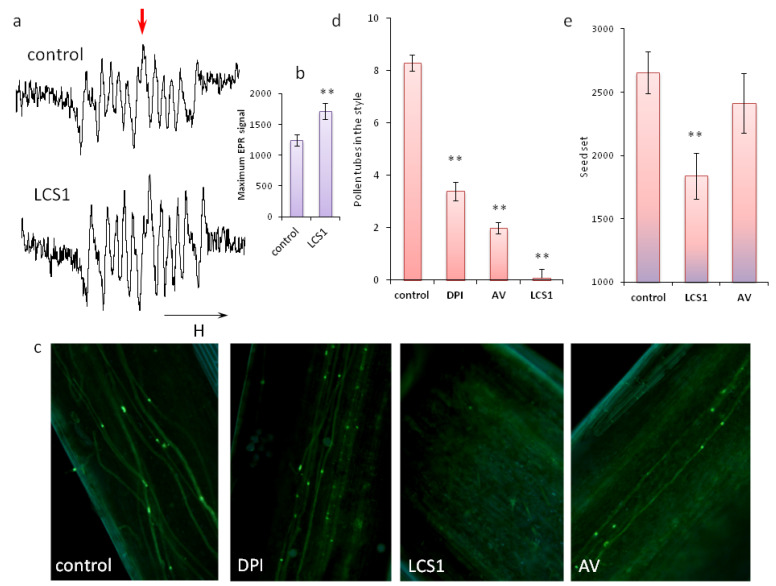
Significance of ROS balance for the rate and efficiency of pollen germination in vivo. (**a**) Typical EPR signals of stigma exudate from control and LCS1-treated flower; (**b**) averaged values of EPR signals (the second peak, indicated by an arrow in (**a**), n = 6); (**c**) typical fluorescent pictures of styles 30 min after pollination (AP), pollen tube wall is stained by decolorized aniline blue, magnification 100×, DPI (diphenyleneiodonium chloride) and AV (acetovanillone): inhibitors of NADPH oxidase, LCS1: inhibitor of Cu/Zn SOD; (**d**) average number of pollen tubes in a pistil style 30 min AP; (**e**) seed set (an average number of seeds in a capsule) 4 weeks AP. Asterisks indicate a significant difference between the means (**- *p* <0,01).

**Figure 4 plants-11-00993-f004:**
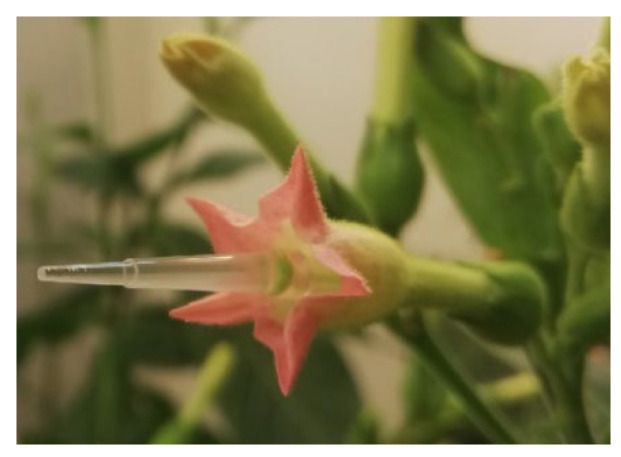
Emasculated tobacco flower at stage 2 with a tip that collects exudate.

## Data Availability

The data presented in this study are available on request from the corresponding author. The data are not publicly available due to the University restrictions related to the educational process.

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
