# Peer review of "The Balance between Different ROS on Tobacco Stigma during Flowering and Its Role in Pollen Germination"

_plants, 2022, doi:10.3390/plants11070993_

Round 1

Reviewer 1 Report

The research entitled “The balance between different ROS on tobacco stigma during flowering and its role in pollen germination” is well conducted and the findings have been well written and discussed. However, certain observations still require some clarification and here are my specific queries and comments regarding that:

  1. In Fig. 1b, do the lower values of EPR signals (corresponding to superoxide level) noted at 3rd and 4th stages of pistil development are statistically significant in comparison to the 2nd stage value? From the graph it appears that 3rd stage EPR is significantly less than 2nd stage EPS but 4th stage EPS is not significantly less than 2nd stage EPS. Please check and revise accordingly in the text and also depict significance levels with */ ** over the graphs (1 b,c and 3 b,d,e).

  1. In line 146, revise ‘Fig. 2c’ as ‘Fig. 2d’.

  1. Line no 92 please rephrase “with was performed 92 on intact flowers”

  1. As per the observations, both AV (inhibitor of NADPH oxidase) and LCS1 (inhibitor of Cu/Zn SOD) treatments significantly lowered pollen germination but the effect of LCS1 on pollen germination was more pronounced in comparison to AV. AV treatment is expected to decrease superoxide level and LCS1 treatment is expected to increase superoxide level (due to non-conversion to hydrogen peroxide) in comparison to the control condition. However in both the cases, hydrogen peroxide production is expected to be less which can disrupt pollen germination. In view of this, I would suggest authors to discuss which one is more critical to lower pollen germination: (a) lesser superoxide production leading to lesser hydrogen peroxide accumulation or (b) lesser conversion of superoxide to hydrogen peroxide.

  1. Generally, upon environmental stresses, ROS level increases in the plant cells which disrupts plant’s physiology and metabolic processes ultimately leading to lower yield. I suggest the authors to briefly discuss the effect of abiotic stress on ROS dynamics and their influence on pollen germination and seed yield.

Author Response

Thank you very much for your valuable comments. We have improved the manuscript, taking into account all the questions that have arisen.
1. We checked the statistical differences, indicated them on the graphs and in the text. It's good that you noticed! We always do it, we just forgot by accident.
2,3. - made changes
4. We changed the text of the discussion, shifting the focus to comparing the effects of inhibitors. However, as a matter of speculation, we put forward a hypothesis explaining the absence of a total effect when blocking NADPH oxidase. However, we will try to test it experimentally in the next study.
5. We have added a new small discussion section on abiotic stress with relevant links. We did not write about this in detail, since no stress was applied in this work.

Reviewer 2 Report

The present study entitled ''The balance between different ROS on tobacco stigma during flowering and its role in pollen germination'' is a well-structured article.  However, a few changes are recommended before it is published.

Comments

1-As already recent investigations are published on the topic so please clarify the novelty of study and you have missed citing recently updated publications on the topic i.e.,

  • Xie, Dong-Ling et al. “Functions of Redox Signaling in Pollen Development and Stress Response.” Antioxidants (Basel, Switzerland) 11,2 287. 30 Jan. 2022, doi:10.3390/antiox11020287
  • Kiyono, H., Katano, K., & Suzuki, N. (2021). Links between Regulatory Systems of ROS and Carbohydrates in Reproductive Development. Plants (Basel, Switzerland)10(8), 1652. https://doi.org/10.3390/plants10081652

Please explain how this work is different from already published work and give reference to these recently published advances on the topic.

2-Please avoid citation of too old data as I have seen citations of the 1960s so add recent development on the topic and avoid too old information.

3-Figure 2 seems blurred, please add a clear picture, I suggest outlining the blots and labeling the groups outside the blot instead of using * over the blots.

Please make sure all the references are including the name of the journal.

Regards

Author Response

Thank you very much for the recommendations, we have included the new review articles suggested by the Reviewer and some of the relevant experimental articles that were cited in them.
We also removed references to some very old works, but still left some to show the historical perspective of these studies.
Scans of gels with SOD activity do not actually look very nice, but this does not reduce their significance. We tried to darken the part where there is no activity, and also followed the advice of the Reviewer: we added frames, outlined the activity bands and signed them next to the gel.
We checked the references (they were generated automatically by the "Mendeley" program), the titles of the journals are present in all cases where applicable.